# Chatbot-based serious games: A useful tool for training medical students? A randomized controlled trial

**Salma Al Kahf[1], Baptiste Roux[2], Sebastien Clerc[1,3], Mona Bassehila[1], A. Lecomte[2], Elsa Moncomble[3], Elodie Alabadan[3], Nina de Montmolin[3], Eve Jablon[1,4], Emilie François[1,5], Gérard Friedlander[1,5], Cécile Badoual[1], Guy Meyer[1,3], Nicolas Roche[1,6], Clémence Martin[1,6☯], Benjamin Planquette ᴵᴰ[1,3☯]***

1 Faculté de Santé, Université de Paris Cité, Paris, France, 2 Fast 4 -Medgame, Nîmes Cedex, France, 3 Service de pneumologie et de soins intensifs, Hôpital Européen Georges Pompidou, AP-HP, Paris, France, 4 Accompagnement à la Gestion de l'Innovation pour la Réussite des étudiants, Univeristé de Paris, Paris, France, 5 Fondation "Sauver la vie", Paris, France, 6 Service de pneumologie, Hôpital Cochin, AP-HP, Paris, France

☯ These authors contributed equally to this work.
* benjamin.planquette@aphp.fr

**Data Availability Statement:** All relevant data are within the paper and its Supporting Information files.

## Abstract

### Objectives

Chatbots, conversational agents that walk medical students (MS) though a clinical case, are serious games that seem to be appreciated by MS. Their impact on MS's performance in exams however was not yet evaluated. Chatprogress is a chatbot-based game developed at Paris Descartes University. It contains 8 pulmonology cases with step-by-step answers delivered with pedagogical comments. The CHATPROGRESS study aimed to evaluate the impact of Chatprogress on students' success rate in their end-term exams.

### Methods

We conducted a post-test randomized controlled trial held on all fourth-year MS at Paris Descartes University. All MS were asked to follow the University's regular lectures, and half of them were randomly given access to Chatprogress. At the end of the term, medical students were evaluated on pulmonology, cardiology and critical care medicine.

### Main outcomes measures

The primary aim was to evaluate an increase in scores in the pulmonology sub-test for students who had access to Chatprogress, compared to those who didn't. Secondary aims were to evaluate an increase in scores in the overall test (Pulmonology, Cardiology and Critical care medicine test (PCC)) and to evaluate the correlation between access to Chatprogress and overall test score. Finally, students' satisfaction was assessed using a survey.

**Funding:** The CHATPROGRESS trial was held in partnership with "Accompagnement à la Gestion de l'Innovation pour la Réussite des étudiants – AGIR" and was funded by the 2018 academic grant "Sauver la vie" from Paris Descartes' Foundation. The funders had no role in study design, data collection and analysis, decision to publish, or preparation of the manuscript.

**Competing interests:** The authors have declared that no competing interests exist.

## Results

From 10/2018 to 06/2019, 171 students had access to Chatprogress (the Gamers) and among them, 104 ended up using it (the Users). Gamers and Users were compared to 255 Controls with no access to Chatprogress. Differences in scores on the pulmonology sub-test over the academic year were significantly higher among Gamers and Users vs Controls (mean score: 12.7/20 vs 12.0/20, p = 0.0104 and mean score: 12.7/20 vs 12.0/20, p = 0.0365 respectively). This significant difference was present as well in the overall PCC test scores: (mean score: 12.5/20 vs 12.1/20, p = 0.0285 and 12.6/20 vs 12.1/20, p = 0.0355 respectively). Although no significant correlation was found between the pulmonology sub-test's scores and MS's assid;uity parameters (number of finished games among the 8 proposed to Users and number of times a User finished a game), there was a trend to a better correlation when users were evaluated on a subject covered by Chatprogress. MS were also found to be fans of this teaching tool, asking for more pedagogical comments even when they got the questions right.

## Conclusion

This randomised controlled trial is the first to demonstrate a significant improvement in students' results (in both the pulmonology subtest and the overall PCC exam) when they had access to Chatbots, and even more so when they actually used it.

## Introduction

Developing an effective teaching strategy for students' training is a common goal to many teachers, especially entertainment-based ones that are gaining in popularity among students. Among those teaching strategies, automated clinical vignettes are tools that can easily be used in medical training. A multicentre American study showed that case-vignettes had fairly similar results to clinical practice audits when it came to quality of care [1].

Recent advances in artificial intelligence helped turning clinical vignettes into handy teaching tools, through chatbots. Chatbots are conversational agents, robots, that walk students though a clinical case. Students engage in a conversation by text or visual communication to conduct clinical and paraclinical examinations. They are then asked to diagnose and devise management plans and see their impact in real-time, all in different settings (clinic, emergency department, a classical hospital ward, an intensive care unit, etc) [2]. Chatbots were also developed for families', to help them with the management of an acute exacerbation of a child's asthma [3]. In a hospital setting, chatbots were used during the COVID-19 pandemic to help screen health-care providers for COVID-19 symptoms and exposures prior to every shift, thereby reducing wait-times and physical proximity with others [4]. In an educational setting, chatbots are still being tested. A recent systematic review of the prior research related to the use of Chatbots in education was performed [5] and points out several important findings. The number of documents relating to the use of Chatbots in education exponentially increased in the recent years, reflecting the current worldwide dynamic to modernize education. The benefits found to chatbots were their quick access to information and more importantly feedback, increasing students' ability to integrate information. When surveyed, medical students claimed to find chatbots appealing based on the recognition, the anthropomorphism in

communication and knowledge expertise of the tool [6]. In nursing students, a knowledge-based chatbot system was found to enhance students' academic performance and learning satisfaction [7]. However, their impact on test-taking was not studied previously in medical students. Their efficacy was not thoroughly tested, due to probably a limited number of examples of chatbots in the European Healthcare curricula [8]. In our trial, we close this gap and aim to evaluate, on fourth-year medical students at Paris Descartes University, the benefit of chatbots' on students' success rate in their end-term exam, as well as their level of satisfaction.

## Methods and materials

### Design and setting

We performed a single-centre open post-test randomized controlled study that compared the overall test scores of students with no access to Chatprogress (Controls) to those of students with access (Gamers) and, among the Gamers, the students who actually used the platform (Users). All fourth-year medical students from Paris Descartes University were eligible. There were no exclusion criteria.

### Study population

During their academic year, fourth-year medical students at Paris Descartes University are divided into 3 groups. They all must go through 3 courses, each consisting of 3 medical specialties, to be taken in a different order by each group throughout the year. Each course lasts 3 months (October to December 2018 –January to March 2019 and April to June 2019). During each course, students have 3 hours/week of lecture per specialty, in a standard classroom setting. At the end of each course, students must take a test on the 3 medical specialties learned during the course (3 sub-tests). Each sub-test is composed of a main clinical case and a few random multiple-choice questions. The course studied during our trial covered pulmonology, cardiology and critical care medicine (PCC). Chatprogress was developed by pulmonologists (CM and BP), it therefore more specifically aimed to train students for the pulmonology sub-test. The clinical cases of the pulmonology exam during the 2018–2019 academic year revolved around sarcoidosis in December 2018, Chronic Obstructive Pulmonary Disease (COPD) and pneumothorax in March 2019, and non-small-cell lung carcinoma in June 2019.

Fourth-year medical students from Paris Descartes University were randomized (1:1). Randomization was done for each group of students going through their PCC course with a computer-generated sequence and a one-to-one ratio (SC). All students were asked to follow their usual timetable as per standard of teaching, irrespective of group assignment. Students randomized in the trial group received an email regarding the study, with the date of an introductory meeting, conducted by BP, CM and BR, to receive further information about Chatprogress. Personal access to the platform was given that day to be used 6 weeks before the exam. Students were asked not to share their Chatprogress access, since masking of students was not feasible because of the nature of the intervention. To incite students to use the Chatbot, the five students with the most amount of finished rounds were offered movie tickets. Students initially randomly assigned to play but who were not given access to Chatprogress because absent at the introductory meeting were added to the control group. Students absent at the end-term exam were excluded from analysis.

### Chatprogress

"Chatprogress" is a website that students can log into to access several chatbot games. Upon logging in, students received an introductory message to explain the trial (Fig 1A, S1 Table).

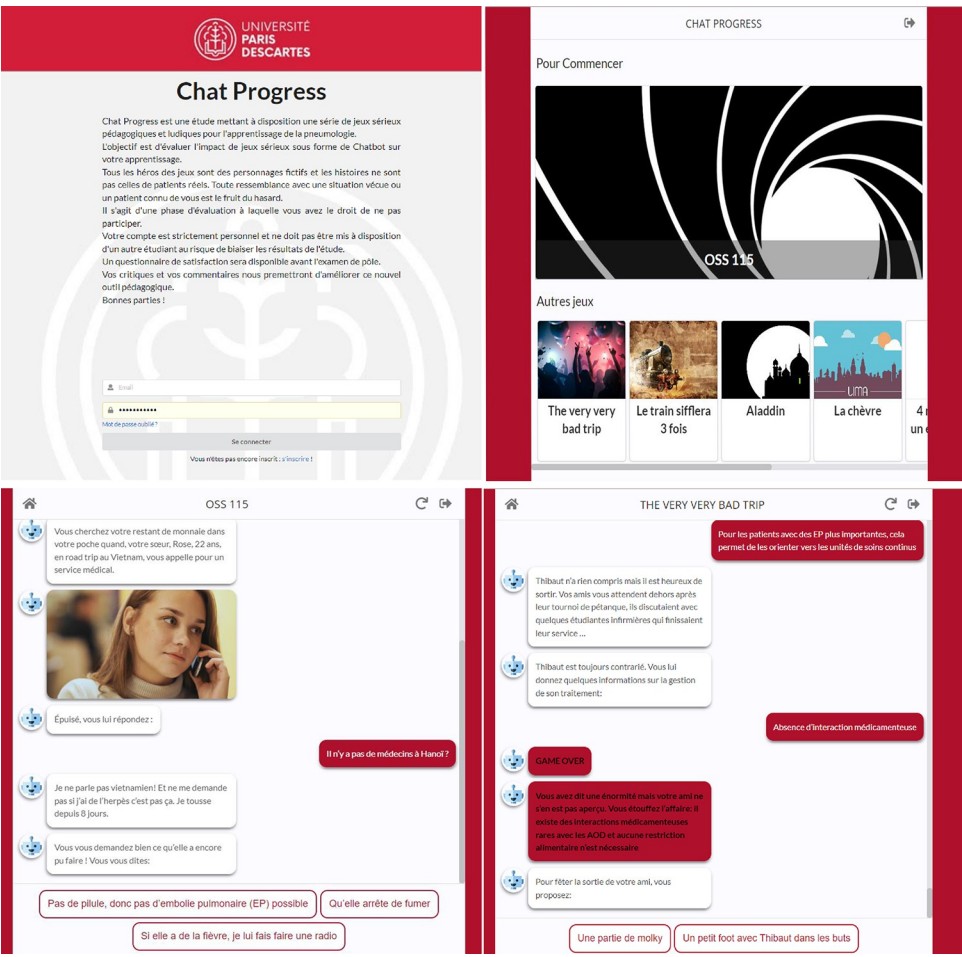

**Fig 1. The chatbot's presentation.** A. Introductory message upon logging in, translated in S1 Table. B. The series of serious games to pick from. C. An example of an interaction with the chatbot, leading to a multiple-choice question. D. An example of a question answered incorrectly, leading to a game over.

Chatprogress design is similar to a streaming platform (Fig 1B). Once a game is chosen, the robot asks a series of questions to walk the student through a medical clinical case while offering multiple answers (Fig 1C). If the student picks a wrong answer, the game is over (GO for Game Over), and a message pops out with a short and funny explanation of why the answer is wrong (Fig 1D). When appropriate, this message is accompanied by a reference from the students' pulmonology textbook [9]. Students can then go back to the game and continue from where they left off or start over. They can also play several rounds of the same game.

The technical aspect of the games was handled by FAST4 company (BR, AL). There were 8 games in total. The clinical cases revolved around 4 major topics in pulmonology (COPD, asthma, pulmonary embolism, and community-acquired pneumonia) for 6 of the games and 3 secondary topics (haemoptysis, pneumothorax and tuberculosis) for the remaining two. The cases were written by BP and CM using youngsters' vocabulary to increase students' engagement. The authors of the games were not involved in the writing of the exam questions and were blinded to the topic of the exam. The games were double-checked by the University's teaching committee (GF, CB, NR) and were tested on 2 students of an upper grade (SA, MB) and 3 pulmonology residents (EM, EA, NDM).

## Collection of information

Information regarding students' interactions with the robot (number of games played, of games finished, of GO and of rounds played) was collected straight from the website by MED-GAME, and the students' exam results (overall and sub-tests' results) were obtained from Paris Descartes University at the end of the academic year. A satisfaction survey was sent out to all students in July of 2019 (S2 Table), regardless of when their PCC exam took place. The survey was only sent out once. No reminder to complete the survey was sent out and no particular strategy was implemented to encourage students to answer the questionnaire. Students had 2 months to answer the survey. The survey was built for the study, asking students 14 questions about Chatprogress. Questions revolved around students' satisfaction of the format and the content of the games. It also interrogated the students on the way they decided to use the games and if they found them useful for their medication education. All but one question were closed questions. The last question was an open question, asking students for feedback to improve the tool.

Results were analysed by SC and CM, blinded to students' group. The chronology of the trial is presented in S1 Fig.

## Objectives and endpoints

The primary objective was to show an improvement in the grades of the pulmonology sub-test. Secondary aims were to demonstrate an increase in scores in the overall PCC test and to evaluate the correlation between access to Chatprogress and overall test score. Assiduity or participation was evaluated by the total number of games started, games finished, and rounds finished by students when students played several rounds of a single game. Finally, students' satisfaction was assessed using a survey.

## Statistical analysis

Given that the distribution of the variables was not Gaussian, differences between exam scores in Controls versus Gamers and versus Users were analysed using a Mann-Whitney U test. A significant difference was defined by a p value $< 0.05$. Correlation analysis between assiduity to Chatprogress and exam scores among Gamers and Users was performed using Spearman rank correlation non-parametric test

All statistical tests were performed using the Prism Software.

## Ethics and participation

The CHATPROGRESS trial was designed by BP and BR and the protocol was approved by Paris Descartes University's teaching committee (GF, CB, GM, NR); students were free to refuse to participate to the study. Consent was given orally.

## Results

### Study population

Between October 1st, 2018 and June 30th, 2019, all 426 fourth-year medical students at Paris Descartes University were randomized to have access or not to the chatbot. Out of the 213 students who were randomized to have access to the chatbot, 42 did not show up at the introductory meeting, therefore were refused access to Chatprogress and consequently assigned to the Control Group. The other 171 students of that group had personal access to the chatbot and were considered the "Gamers". Out of the 171 gamers, 104 students eventually logged-in and constituted the User's Group. The remaining 67 never logged-in and did not play a single

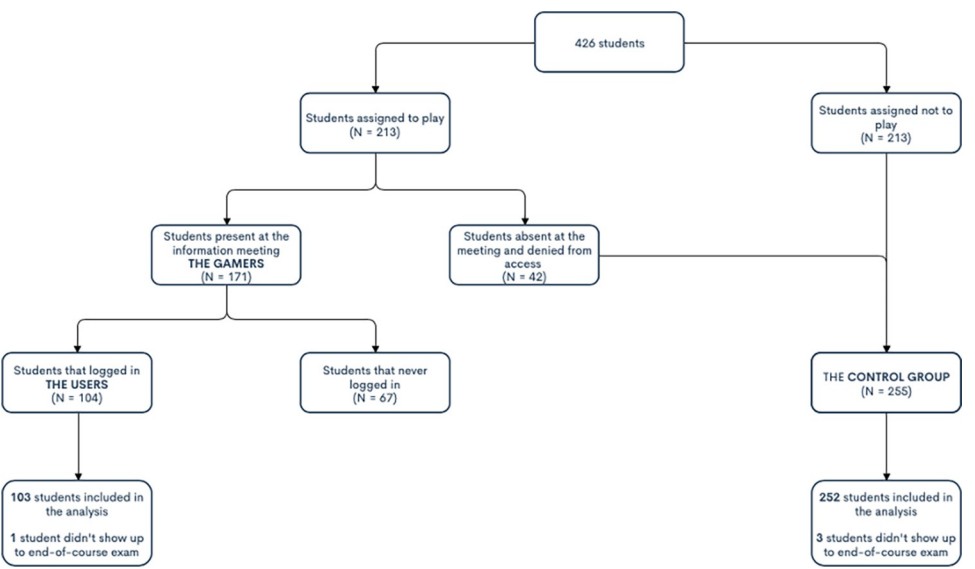

**Fig 2. The Chatprogress trial's flow chart.**

game. We were thus left with 171 Gamers, 104 Users and 255 students in the control group. One user and 3 students in the control group did not show up to the end-of-course exam. These students were excluded from the final analysis (Fig 2).

The baseline characteristics, in terms of sex ratio and age, did not differ between both groups.

The users ended up playing 5105 rounds. On average per user, 48.8 (± 33.9) rounds were played by each user, 6.6 (± 4.3) rounds were finished, 4.6 (± 1.9) games were finished, and each user got a GO 41.9 (± 30.6) times. Two games, games number 2 and 5, revolving around community-acquired pneumonia and COPD respectively, were never finished (Table 1).

## Results of the pulmonology and the overall PCC exams

The primary outcome was assessed in the 422 students that showed up to the exam. The Gamers and Users had significantly better results in their pulmonology subtest with a mean score (± SD) of 12.7/20 ± 2.8, p = 0.0104 (Median difference of 1.2; 95% IC: 0.17 to 1.33) and 12.7/20 ± 2.7, p = 0.0365 (Median difference of 1.2; 95% IC: 0.03 to 1.42) respectively, versus the Controls: 12.0/20 ± 2.9 (Fig 3A). The same result was observed in the overall PCC exam results, Gamers and Users having significantly higher grades with a mean score (± SD) of 12.5/20 ± 1.8, p = 0.0285 (Median difference of 0.45; 95% IC: 0.425 to 0.775) and 12.62/20 ± 1.7, p = 0.0355 (Median difference of 0.46; 95% IC: 0.03 to 0.88) respectively versus vs Controls: 12.1/20 ± 1.9) (Fig 3B).

**Table 1. Data collected by MedGame on the use of the platform.**

| Game n˚ | I | II | III | IV | V | VI | VII | VIII | Total |
|---|---|---|---|---|---|---|---|---|---|
| Topic | Asthma | CAP | PE | PTX | COPD | COPD | TB/H | PE | |
| GO | 581 | 466 | 1493 | 393 | 296 | 406 | 349 | 420 | 4404 |
| W | 184 | 0 | 143 | 117 | 0 | 95 | 87 | 75 | 701 |
| Total | 765 | 466 | 1636 | 510 | 296 | 501 | 436 | 495 | 5105 |

n˚ = number; GO = number of Game Overs; W = number of finished rounds; CAP = community-acquired pneumonia; PE = pulmonary embolism;

PTX = pneumothorax; COPD = chronic obstructive pulmonary disease; TB = tuberculosis; H = haemoptysis.

A. Comparison of grades obtained in pulmonology

B. Comparison of average grades obtained in pooled pulmonology, cardiology and intensive care medicine.

**Fig 3.** A. Comparison of grades obtained in pulmonology. B. Comparison of average grades obtained in pooled pulmonology, cardiology and intensive care medicine (PCC).

## Students' assiduity, performance and satisfaction

We did not find a significant correlation between the pulmonology grades and the total number of games started or finished in the Users' group. No correlation was found between the pulmonology grades and the number of times a User finished a game (i.e., number of rounds finished) either (S2 Fig).

Although not significant, the pulmonology grades of Users tested on a subject covered by Chatprogress (i.e. in the second trimester, tested on COPD and pneumothorax) revealed a trend towards a positive correlation in assiduity parameters (number of games started, finished among the 8 and numbers of times a user finished a game), as shown in Fig 4.

Qualitative evaluation of students' satisfaction was evaluated with a satisfaction survey. Twenty-seven of the 104 Users replied (26%). The results are shown in Table 2. Overall, the students seemed satisfied, from the games' general presentation to their benefit in the students' medical training. Game 5, one of the two games that were never finished, was the most disliked by students. The majority of the students got GOs voluntarily (70%) and played a game again even if they successfully got to the end of it (59%). Most of them considered Chatprogress useful for learning medical concepts (88.8%) or reviewing them (70%). They all (100%) considered Chatprogress as an interesting tool to have on the long run, with more games, covering more of their courses.

## Discussion

Our results showed that using Chatprogress, a chatbot system, potentially improved students' results on an academic test. The study also showed that medical students not only appreciated the chatbots but more importantly used it as a pedagogical tool. In fact, the intensity of the use of the platform shows a trend to correlate with students' results: the more games were played

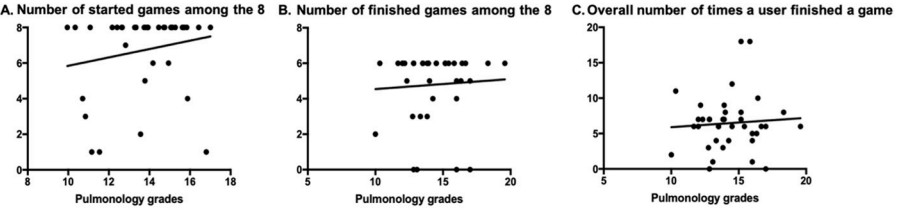

A. Number of started games among the 8    B. Number of finished games among the 8    C. Overall number of times a user finished a game

**Fig 4. Correlations between the grades in pulmonology and students' assiduity parameters when the exam subject was covered by Chatprogress.**

**Table 2. Results[μ] of the students' satisfaction survey.**

| A | | | | | |
|---|---|---|---|---|---|
| | **Students' response** | | | | |
| **Question** | **Great (%, (n))** | **Good (%, (n))** | **Average (%, (n))** | **Useless/No (%, (n))** | **Skipped (%, (n))** |
| What do you think of this game format? | 59% (16) | 33% (9) | 7% (2) | 0% (0) | 0% (0) |
| Do you think the games fit with your training? | 34% (9) | 53% (14) | 11% (3) | 0% (0) | 0% (0) |
| Did you enjoy playing the games? | 40.7% (11) | 55% (15) | 3% (1) | 0% (0) | 0% (0) |

| B | | | | | | | | | | |
|---|---|---|---|---|---|---|---|---|---|---|
| | **Students' response** | | | | | | | | | |
| Which game did you like the most? | **I** | **II** | **III** | **IV** | **V** | **VI** | **VII** | **VIII** | **None** | 5 |
| | 3 | 0 | 8 | 1 | 0 | 0 | 0 | 1 | 9 | |

| C | | | | | | | | | | |
|---|---|---|---|---|---|---|---|---|---|---|
| | **Students' response** | | | | | | | | | |
| Which game did you like the least? | **I** | **II** | **III** | **IV** | **V** | **VI** | **VII** | **VIII** | **None** | 7 |
| | 2 | 0 | 1 | 0 | 5 | 0 | 0 | 0 | 12 | |

| D | | | |
|---|---|---|---|
| | **Students' response** | | |
| | **Yes** | **No** | |
| Did you make anyone else try it out? | 11.1% (3) | 88.8% (24) | 0 |
| Did you get GO voluntarily? | 70% (19) | 29% (8) | 0 |
| If you successfully made it through a game, did you ever play the game again? | 59% (16) | 40% (11) | 0 |
| Was this format useful for learning concepts? | 88.8% (24) | 11.1% (3) | 0 |
| Was this format useful for reviewing concepts? | 70% (19) | 29% (8) | 0 |
| Would you like to have access to such a gaming platform for all your teaching modules, with a big variety of games? | 100% (27) | 0% (0) | 0 |

| E | | | | | |
|---|---|---|---|---|---|
| | **Students' response** | | | | |
| Games' level of difficulty | **Very easy** | **Easy** | **Accessible** | **Hard** | **Very Hard** | 0 |
| | 0% (0) | 14% (4) | 77% (21) | 7% (2) | 0% (0) | |

| F | | | |
|---|---|---|---|
| | **Students' response** | | |
| Games' duration | **Too long** | **Just right** | **Too short** | 0 |
| | 0% | 85% (23) | 15% (4) | |

MCQ = multiple choice questions

μ: responses to questions 2 to 14. Question 1 was an open question asking for students' email addresses and question 15's answers are found in S3 Table.

or finished, and the more rounds were finished, the better the students performed on their pulmonology test when the subject was covered by Chatprogress.

Chatprogress was created to ease the process of learning, using a tool that fits right into the students' schedules, and pockets [10]. Chatbots offer clinical scenarios of different lengths and levels, depending on how much time the student wants to allocate to the game. Games also fulfil students' need for guidance, with step-by-step explanation, all the while still autonomizing them [11]. This is particularly helpful with medical education's continuity in times when in-person teaching is not feasible, such as during the Covid-19 pandemic [12]. This tool can replace the often crowded and not always accessible simulation labs, and preserves the lab's sense of active interaction rather than the passive transmission of information. In fact, a study

showed that a serious game on a mannequin was not superior to when the same scenario was played out on a computer in a gaming format to train medical students on the management of a cardiac arrest [13]. Chatbots also help students work on 4 of the 6 components of medical reasoning such as information gathering, hypothesis generation or management and treatment [14]. Finally, the gaming aspect of the tool, with a reward system or a multiplayer mode, also appeals to youngsters, making the tool not only handy but also fun.

Most of the learning on the platform is done through trial and error, a well-known and efficient learning method [15]. Students feel more comfortable making mistakes and learning from them when medicine is practiced on a robot, rather than a real patient, and when these mistakes are made outside of the monitoring environment of a teaching hospital [16].

Students were found to really appreciate this mean of teaching. This was concluded not only from the answers of the survey but was also proven by the students' voluntary GOs to receive extra explanations. This suggests that students were not merely guessing their answers but were genuinely thinking the questions through. It also justifies students' request to add explanations even when they got a question right to make sure their clinical reasoning held up. Furthermore, part of the games' strength relies on its ability to teach medical reasoning by repetition, a powerful learning technique [17]. This feature of the games was well utilized by students: according to the survey, 59% of students who successfully made it to the end of a game played that game again. The games were also considered of a decent level of difficulty, making Chatprogress a handy tool not only to revise concepts but also to learn them right off the bat. As the survey shows that two games could not be completed, we think that a debriefing session would allow us to understand the difficulties either inherent to the chatbot or the difficulties of medical reasoning encountered by the students and, thus, to correct them.

We made sure, to the best of our ability, to limit the cross-over between groups, by first only meeting up with the students that were randomized in the trial group. Personal identification was also distributed, limiting the access of students in the control group to the tool. We also made sure the group contamination was limited by asking the question in the survey, which pointed out that cross-over was limited among those who answered the survey. However, we cannot eliminate a non-response bias, remaining ignorant of the sharing of the platform by those who did not fill out the survey. We also cannot argue that some of the knowledge acquired by using the chatbot could have been shared between peers.

Our trial however faces an important limitation: the relatively weak adherence of students to the trial and the survey. First, a selection bias could result from students not showing up the meeting, moving them to the control group in our per-protocol analysis. Students who thus finally constituted the Gamers' group were more likely to be more assiduous than those in the control group. Similarly, only 26% of users completed the survey. Thus, we can't rule out that those who answered were those who were particularly fond of the concept. Likewise, grades were not statistically higher in Users when the test revolved around a subject covered by Chatprogress probably because of the small number of students of that trimester (n = 38).

An explanation to students' weak adherence to the trial could be that, in Paris Descartes, the PCC course is taught during their first clinical year. Students during that year learn to juggle between hospital obligations, lectures, study groups and self-education. That could have hindered the introduction of yet another learning tool. Students' poor response rate to the survey can be explained partly by the timing of when it was sent out. The email was sent at the end of the year, so 3 to 6 months after the completion of the PCC exam for two thirds of the students. It was sent out in July and students had 2 months to fill it out. It was therefore sent during the summer break, when students are usually the least responsive. This is also why a second blast, a few weeks later, was not sent, although a pre-notification or a subsequent reminder after the initial survey would have increased the response rate [18]. An incentive was

also not considered, although was it was also shown to be linked to an increased response rate, with the budget entirely dedicated to the development of the chatbot and the student's incentive to use it.

Our study was also a single centre one. It was performed on a single class with a limited number of students, targeted a single course, with a limited number of clinical scenarios, that all revolved around only one of the modules, pulmonology. Another limitation could be the use of only multiple-choice questions, which do not reflect how real-life conversations with patients are held. This is a disadvantage that our chatbot has compared to a scenario on a mannequin. We thus wish to develop games in the future with open questions. In addition, we understand that students' ability to pass an exam is multifactorial, going from students' schedule, to participation rate, to interest in the field or simply luck on exam day. We tried to minimize the confusion biais by performing a randomized controlled trial, with however a remaining attrition bias, that is hard to take into consideration considering the limited number of students per trimester, and evaluation bias on the evaluation of the primary outcome with students of different trimesters being evaluated on different topics.

Similarly, another miss-match with reality would be our chosen outcome, grades on a standardized test, rather than students' performance in the hospital, the ultimate goal of medical education. However, it was shown that clinical competency assessments, such as the pulmonology test at Paris Descartes that is built around a clinical case, are strong predictors of internship performance [19]. In addition, gaging the impact of 6 weeks spent on a gaming platform on performance in a hospital, it being the result of a lengthy and meandering journey, is challenging at best.

## Conclusion

In this single-centre open randomized control trial, the use of chatbot was found to significantly increase students' average score in the specialty covered by the chatbot but also in the overall PCC exam score although No significant correlation was found between students' assiduity to the platform and pulmonology exam results. This is to our knowledge the first randomized controlled trial to study not only students' participation and satisfaction, but also the effect that serious games through chatbots have on their performance on exam day. Chatbots could thus be a potential tool for learning in medicine, where the evaluation of a reasoning takes on full importance.

Futures studies, should multiply the number of games and analyse chatbots' effect on multiple courses, at all levels of medical school.

## Supporting information

**S1 Table. Introductory message upon logging in.**
(DOCX)

**S2 Table. Satisfaction survey.**
(DOCX)

**S3 Table. Responses to question 15 of the satisfaction survey.**
(DOCX)

**S1 Fig. Chronology of the CHATPROGRESS trial during the 2018–2019 school year.** 6w: six weeks.
(TIF)

**S2 Fig. Correlations between students' assiduity parameters and pulmonology grades among all users, n = 103.**
(TIF)

## Acknowledgments

The authors would like to thank Jean François Mescoff for his enormous help in bringing this project to life.

Professor Guy Meyer passed away in December 2020. He worked with Doctor Planquette in the early stages of the Chatprogress project. On top of being a skilled clinician and researcher, he was a unique medical teacher and remained involved in Paris Descartes University for as long as his illness allowed. This manuscript is a tribute to his memory.

## Author Contributions

**Conceptualization:** Baptiste Roux, A. Lecomte, Elsa Moncomble, Elodie Alabadan, Nina de Montmolin, Emilie François, Gérard Friedlander, Cécile Badoual, Guy Meyer, Nicolas Roche, Clémence Martin, Benjamin Planquette.

**Data curation:** Baptiste Roux, Guy Meyer, Clémence Martin, Benjamin Planquette.

**Formal analysis:** Sebastien Clerc, Clémence Martin, Benjamin Planquette.

**Funding acquisition:** Eve Jablon, Emilie François, Gérard Friedlander, Cécile Badoual, Guy Meyer, Clémence Martin, Benjamin Planquette.

**Investigation:** Guy Meyer, Clémence Martin, Benjamin Planquette.

**Methodology:** Sebastien Clerc, Clémence Martin, Benjamin Planquette.

**Project administration:** Salma Al Kahf, Mona Bassehila, Eve Jablon, Nicolas Roche, Clémence Martin, Benjamin Planquette.

**Resources:** Benjamin Planquette.

**Software:** Sebastien Clerc, Clémence Martin.

**Supervision:** Clémence Martin, Benjamin Planquette.

**Validation:** Clémence Martin, Benjamin Planquette.

**Visualization:** Clémence Martin, Benjamin Planquette.

**Writing – original draft:** Salma Al Kahf.

**Writing – review & editing:** Salma Al Kahf, Baptiste Roux, Mona Bassehila, Nicolas Roche, Clémence Martin, Benjamin Planquette.

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
