## [Decision Letter · Decision Letter 0]

12 Jul 2022

PONE-D-22-00159Chatbot-based serious games: a useful tool for training medical students? A randomized controlled trial.PLOS ONE

Dear Dr. Planquette,

Thank you for submitting your manuscript to PLOS ONE. After careful consideration, we feel that it has merit but does not fully meet PLOS ONE’s publication criteria as it currently stands. Therefore, we invite you to submit a revised version of the manuscript that addresses the points raised during the review process.

We look forward to receiving your revised manuscript.

Kind regards,

Elsayed Abdelkreem, MD, PhD

Academic Editor

PLOS ONE

“The CHATPROGRESS trial was designed by BP and BR and the protocol was approved by Paris Descartes University’s teaching committee (GF, CB, GM, NR); students were free to refuse to participate to the study. The CHATPROGRESS trial was held in partnership with “Accompagnement à la Gestion de l’Innovation pour la Réussite des étudiants – AGIR" (EJ) and was funded by the 2018 academic grant “Sauver la vie” from Paris Descartes’ Foundation (ES, GF).”

“The CHATPROGRESS trial was held in partnership with “Accompagnement à la Gestion de l’Innovation pour la Réussite des étudiants – AGIR" and was funded by the 2018 academic grant “Sauver la vie” from Paris Descartes’ Foundation. The funders had no role in study design, data collection and analysis, decision to publish, or preparation of the manuscript.”

3. PLOS requires an ORCID iD for the corresponding author in Editorial Manager on papers submitted after December 6th, 2016. Please ensure that you have an ORCID iD and that it is validated in Editorial Manager. To do this, go to ‘Update my Information’ (in the upper left-hand corner of the main menu), and click on the Fetch/Validate link next to the ORCID field. This will take you to the ORCID site and allow you to create a new iD or authenticate a pre-existing iD in Editorial Manager. Please see the following video for instructions on linking an ORCID iD to your Editorial Manager account: https://www.youtube.com/watch?v=_xcclfuvtxQ.

6. Please include your tables as part of your main manuscript and remove the individual files. Please note that supplementary tables (should remain/ be uploaded) as separate ""supporting information"" files.

Reviewers' comments:

Reviewer's Responses to Questions

**Comments to the Author**

1. Is the manuscript technically sound, and do the data support the conclusions?

Reviewer #1: Yes

Reviewer #2: Partly

2. Has the statistical analysis been performed appropriately and rigorously? 

Reviewer #1: Yes

Reviewer #2: No

3. Have the authors made all data underlying the findings in their manuscript fully available?

Reviewer #1: Yes

Reviewer #2: Yes

4. Is the manuscript presented in an intelligible fashion and written in standard English?

Reviewer #1: Yes

Reviewer #2: Yes

5. Review Comments to the Author

Reviewer #1: The authors implemented a new teaching platform related to the game-based learning method in teaching medicine which is a good tool that is of help in the engagement of the students in the learning process.

Introduction: is satisfactory

Aim of the work: The authors mentioned that their aim statement aims to evaluate the benefit of chatbots’ on students’ success rate in their end-term exams, as well as their level of satisfaction.

Methods

Study population:

They implemented their trial on two batches of students; It is better to compare the achievement of the two baches to each other and evaluate their responses separately to evaluate the program's durability on two different occasions. Moreover, It was better to implement more than one course regarding the same bach.

Chatprogess program: it is better to mention the link to the program's free trial to be tested by the readers.

The authors should include the component of the satisfaction survey in detail and mention if is it validated or not and the method of validation.

Results: are satisfactory

Discussion: satisfactory with minor comments. The authors should discuss why not all assigned students are not involved in the games and explain the cause of their withdrawal. Moreover, explain the low percentage of students who came to fill the survey

Reviewer #2: Thank you for the opportunity to review the manuscript. I found that the study is interesting. Hence, I have several comments to further improve the quality of the paper.

Introduction

1. “Developing new teaching strategies…”. I don’t think that this is the common goal as we (the educators) do not want to flood the practice with various strategies. Supposedly “developing an effective teaching strategies…”

2. “Robotized clinical vignettes…” is it using robot? There is a different between robotized and AI-computerized/AI-powered/automated, which in the case of the citation you provide, not using robot. As there are several interpretations on what robot is. For a safer term, perhaps you can use “technology-application clinical vignettes”.

3. “an intensive care unit…” use “etc.” rather than “…”

4. The introduction is insufficient. There argument provided is less robust. The authors need to include more literature and provide argument to justify the needs of the current study. Example of literature should be included:

• Chang, C.-Y., Kuo, S.-Y., & Hwang, G.-H. (2022). Chatbot-facilitated Nursing Education: Incorporating a Knowledge-Based Chatbot System into a Nursing Training Program. Educational Technology & Society, 25(1), 15–27. https://www.jstor.org/stable/48647027

• Okonkwo, C.W. & Ade-Ibijola, A. (2021). Chatbots applications in education: A systematic review, Computers and Education: Artificial Intelligence, 2, 100033. https://doi.org/10.1016/j.caeai.2021.100033

• Frangoudes et al. (2021). An Overview of the Use of Chatbots in Medical and Healthcare Education. Learning and Collaboration Technologies: Games and Virtual Environments for Learning: 8th International Conference, 170–184. https://doi.org/10.1007/978-3-030-77943-6_11

Methods

1. You need to explicitly mention that your study is only post-test of RCT.

2. Explanation on how the students were randomized into group needs to be moved from data analysis to study population (where you mentioned about randomization) section.

3. There is no description on the satisfaction survey. The author should explain either the survey is standardized or build by the researcher purposely for the study (if purposely build, then need to tell how it was developed; by discussion? Based on previous literature review?, and how the validity of the questionnaire is ensured), tell on how many questions available, assess on what (the domains), how to answer each question (by Likert scale, dichotomous Yes/No answer), and how long the duration required to complete the questionnaire.

4. Is there any strategy use to encourage the experiment group to utilize the chatbot? Please describe.

5. What strategy you use to ensure optimal response for your participants to answer the survey? When the survey given? Is there reminder provided? How many blasts? How long the duration given for them to complete the survey?

6. “Information regarding students’ interactions with the robot…” please change robot to chatbot or game.

7. Why you use Mann-Whitney U test instead of Independent t-test? Please provide justification to support.

8. Please refer to the CONSORT checklist when preparing for your revised manuscript to ensure the comprehensiveness of information.

Result

1. The 171 samples in the experiment group consist of only 104 using the chatbot and the remaining of 67 never use it. So, you cannot analyze the outcome using the 171. The 67 is considered as dropout and cannot be included in the final analysis. The reason is the 67 samples never access to the chatbot and considered no different than those in control group. It is either you move the 67 to control group or exclude them from the analysis. If possible, please use CONSORT flowchart for your figure 2.

2. Again, your analysis on the PCC exam results needs to consider the above point I argued.

3. The difference although significant but small. You need to conduct additional statistical analysis such as effect size and minimal detectable change/minimum clinically important analysis to support your findings.

Discussion

1. You mentioned the limitation on the low adherence, but there is no direct discussion about why there is a significant number of students never use the chatbot.

2. You mentioned the limitation on the low response rate but there is no discussion about why on the low response rate on the survey.

3. “Our results showed that using Chatprogress, a chatbot system, improved students’ results on an academic test.” I less agree with the statement as it needs to be cautiously accepted. Although significant, but the visual analysis (mean comparison) shows little difference and no significant correlation between the frequency of chatbot use with grades. I suggest changing to a more cautious tone such as “potentially improved” or “plausible in improving”.

4. More aspect on the feasibility can be discussed as the survey findings yielded several interesting issues. For example, the same game (e.g., I, III) is rated in both as like the most and least. Another example is game II and V has no participant complete rounds.

5. Another interesting aspect is high number intentionally get GO (Game Over), is it because they just want to quickly get the answer (information reading) rather than playing the game? You touched about this in brief as “learning on the platform is done through trial and error, a well-known and efficient learning method”. However, a deeper analysis is required. Is it perhaps playing game is time consuming, so they quickly want the answer? So, they can use the save time for something else? Please add more reference and argument.

6. “This suggests that students were not merely guessing their answers but were genuinely thinking the questions through”. I found this argument is weak if want to be based on high number of voluntary GO. It can be that they do not want to think at all and terminate the game to directly get answer/reasoning?

7. I am not sure if cross-over is limited. Even with 15.9% (n=27) response rate, there is an 11% answered YES that they make anyone else try the CHATPROGRESS. There is a significant non-response bias to be considered. Moreover, you cannot sure that they at least not discuss about the information with their friends (not about using the chatbot, but sharing the knowledge they gained from it).

8. “Similarly, only 13% of the Gamers completed the survey”. 13% or 15.9%? as it is contradicted between the one mentioned in the discussion and the one reported in the result.

9. You need to critically discuss that student motivation to pass exam is multifactorial, not just about using the game. Therefore, this may explain of why there is only a small difference between the experiment and control outcome. Moreover, the use of game perhaps gave an opportunity bias as the intensity is not properly controlled and similarly levelled (example, the experiment group has double the time for revision – from book and game). Biases and limitation of using RCT in medical education research need to be properly discussed.

10. If possible, include more reference and provide a deep discussion.

My condolences to your team and family of Professor Guy Meyer. I believe he will be proud with the work.

6. PLOS authors have the option to publish the peer review history of their article (what does this mean?). If published, this will include your full peer review and any attached files.

Reviewer #1: **Yes: **Ayman Z. Elsamanoudy

Reviewer #2: **Yes: **Muhammad Hibatullah Romli

---

## [Author Response · Author response to Decision Letter 0]

23 Oct 2022

Reviewers' comments:

Reviewer #1: 

The authors implemented a new teaching platform related to the game-based learning method in teaching medicine which is a good tool that is of help in the engagement of the students in the learning process.

Introduction: is satisfactory

Aim of the work: The authors mentioned that their aim statement aims to evaluate the benefit of chatbots’ on students’ success rate in their end-term exams, as well as their level of satisfaction.

Methods

Study population:

They implemented their trial on two batches of students; It is better to compare the achievement of the two baches to each other and evaluate their responses separately to evaluate the program's durability on two different occasions. Moreover, It was better to implement more than one course regarding the same bach.

Answer

We thank the reviewer for this interesting comment. We understand that reviewer 1 would have liked to see an analysis” gamers versus controls” for each of the 3 tests, for all 3 medical specialties of the PCC course. This analysis had been done but not proposed in our results section. Gamers and Users had better results in their pulmonology subtest for each trimester, but the difference was not statically significant, whereas the difference between players and controls was significant when compared across the year. We considered that non significance for analysis by trimester was due to lack of power, and thus decided to focus on the main result for the whole study population. 

We decided to limit the chatbots to pulmonology because our funding allowed for the development of 8 games, which is just enough to cover the pneumology program. Expanding the Chatprogress project by adding other courses is what we are looking forward to doing and are currently seeking funding for.

Chatprogess program: it is better to mention the link to the program's free trial to be tested by the readers. 

Answer

We thank the reviewer for his interesting comment and find his proposal relevant. Nevertheless, Chatprogress belongs to the Université de Paris Cité and cannot be made freely available for legal reasons. Moreover, chatbots are only in French for the moment. 

We suggest to the readers of the article to visit the free online platform (https://www.medgame.com/) developed since by the provider FAST4 company. The games, however, are also in French.

We have therefore added the following sentences to the section “disclosure of conflict of interests”: "For more information on Chatprogress, contact the corresponding author (benjamin.planquette@aphp.fr). Free medical chatbots in French are available on the platform: https://www.medgame.com/.”

The authors should include the component of the satisfaction survey in detail and mention if is it validated or not and the method of validation.

Answer

The satisfaction survey is detailed in the “Supplementary material” and extra information regarding the survey was added in Methods, as follows : A satisfaction survey was sent out to all students in July of 2019 (Supplementary table 2), regardless of when their PCC exam took place. The survey was only sent once, no reminder to complete the survey was sent out and no particular strategy was implemented to encourage students to answer the questionnaire. Students had 2 months to answer the survey. The survey was built for the study, asking the students 14 questions about Chatprogress. Questions revolved around students’ satisfaction of the format and the content of the games. It also interrogated the students on the way they decided to use the games and if they found them useful for their medication education. All but one question were closed questions. The last question was an open question, asking students for feedback to improve the tool.”

Results: are satisfactory

Discussion: satisfactory with minor comments. The authors should discuss why not all assigned students are not involved in the games and explain the cause of their withdrawal. Moreover, explain the low percentage of students who came to fill the survey. 

Answer

Thank you for pointing out that we indeed did not explain this. We tried to make up for it in the discussion by adding the following : “An explanation to students’ weak adherence to the trial could be that, in Paris Descartes, the PCC course is taught during their first clinical year. Students during that year learn to juggle between hospital obligations, lectures, study groups and self-education. That could have hindered the introduction of yet another learning tool. Students’ poor response rate to the survey can be explained partly by the timing of when it was sent out. The email was sent at the end of the year, so 3 to 6 months after the completion of the PCC exam for two thirds of the students. It was sent out in July and students had 2 months to fill it out. It was therefore sent during the summer break, when students are usually the least responsive. Also, no reminder was sent out after the initial email.”

 

Reviewer #2:

Thank you for the opportunity to review the manuscript. I found that the study is interesting. Hence, I have several comments to further improve the quality of the paper.

Introduction

1. “Developing new teaching strategies…”. I don’t think that this is the common goal as we (the educators) do not want to flood the practice with various strategies. Supposedly “developing an effective teaching strategies…”

Answer

The introduction has been modified accordingly: “Developing an effective teaching strategy for students’ training is a common goal to many teachers,”

2. “Robotized clinical vignettes…” is it using robot? There is a different between robotized and AI-computerized/AI-powered/automated, which in the case of the citation you provide, not using robot. As there are several interpretations on what robot is. For a safer term, perhaps you can use “technology-application clinical vignettes”.

Answer

We modified our sentence accordingly, by using the term “automated clinical vignettes ».

3. “an intensive care unit…” use “etc.” rather than “…”

Answer

As asked by the reviewer, «… » was removed and replaced by “etc.”

4. The introduction is insufficient. There argument provided is less robust. The authors need to include more literature and provide argument to justify the needs of the current study. Example of literature should be included:

• Chang, C.-Y., Kuo, S.-Y., & Hwang, G.-H. (2022). Chatbot-facilitated Nursing Education: Incorporating a Knowledge-Based Chatbot System into a Nursing Training Program. Educational Technology & Society, 25(1), 15–27. https://www.jstor.org/stable/48647027

• Okonkwo, C.W. & Ade-Ibijola, A. (2021). Chatbots applications in education: A systematic review, Computers and Education: Artificial Intelligence, 2, 100033. https://doi.org/10.1016/j.caeai.2021.100033

• Frangoudes et al. (2021). An Overview of the Use of Chatbots in Medical and Healthcare Education. Learning and Collaboration Technologies: Games and Virtual Environments for Learning: 8th International Conference, 170–184. https://doi.org/10.1007/978-3-030-77943-6_11

Answer

Thank you for the interesting comment. We have modified the introduction accordingly by adding arguments to justify the need for our study, and solidifying the arguments with the example of literature pointed out. We added the following : “A recent systematic review of the prior research related to the use of Chatbots in education was performed (5) and points out several important findings. The number of documents relating to the use of Chatbots in education exponentially increased in the recent years, reflecting the current worldwide dynamic to modernize education. The benefit of chatbots were their quick access to information and more importantly feedback, increasing students’ ability to integrate information. (…) In nursing students, a knowledge-based chatbot system was found to enhanced students’ academic performance and learning satisfaction (7). However, their impact on test-taking was not studied previously in medical students. Their efficacy was not thoroughly tested, due to probably a limited number of examples of chatbots in the European Healthcare curricula (8).”

Methods

1. You need to explicitly mention that your study is only post-test of RCT.

Answer

We thank the reviewer for this interesting methodological comment and we included the mention by adding it in the abstract and the Methods : “We performed a single-centre open post-test randomized controlled study“.

2. Explanation on how the students were randomized into group needs to be moved from data analysis to study population (where you mentioned about randomization) section.

Answer

All the information regarding randomization of students was moved to the « Study population » section of Methods.

3. There is no description on the satisfaction survey. The author should explain either the survey is standardized or build by the researcher purposely for the study (if purposely build, then need to tell how it was developed; by discussion? Based on previous literature review?, and how the validity of the questionnaire is ensured), tell on how many questions available, assess on what (the domains), how to answer each question (by Likert scale, dichotomous Yes/No answer), and how long the duration required to complete the questionnaire.

Answer

The survey itself is detailed in the Supplementary Material, and further information regarding the survey is now developed in the « Collection of information » section of Methods, as follows: “The survey was only sent out once, no reminder to complete the survey was sent out and no particular strategy was implemented to encourage students to answer the questionnaire. Students had 2 months to answer the survey. The survey was built for the study, asking the students 14 questions about Chatprogress. Questions revolved around students’ satisfaction of the format and the content of the games. It also interrogated the students on the way they decided to use the games and if they found them useful for their medication education. All but one question were closed questions. The last question was an open question, asking students for feedback to improve the tool.”

4. Is there any strategy use to encourage the experiment group to utilize the chatbot? Please describe.

Answer

Indeed, students with the most amount of finished rounds were offered movie tickets. We added that information in the methods section, as follows: “To incite students to use the Chatbot, the five students with the most amount of finished rounds were offered movie tickets”.

5. What strategy you use to ensure optimal response for your participants to answer the survey? When the survey given? Is there reminder provided? How many blasts? How long the duration given for them to complete the survey?

Answer

Thank you again for pointing out that the paper indeed lacked information regarding the survey. We mentioned all of the missing information in the Methods section, as explained in question 3 (see above).

6. “Information regarding students’ interactions with the robot…” please change robot to chatbot or game.

Answer

Thank you, and so we did : “Information regarding students’ interactions with the chatbot (number of games played, of games finished, of GO and of rounds played) was collected….”

7. Why you use Mann-Whitney U test instead of Independent t-test? Please provide justification to support.

Answer

We chose this test because the distribution of the variables was not Gaussian. 

We aimed at comparing two independent samples with an outcome not normally distributed, therefore a nonparametric Mann-Whitney U test was used. This information has been added in the revised manuscript as follows : “ Given that the distribution of the variables was not Gaussian, differences between exam scores in Controls versus Gamers and versus Users were analysed using a Mann-Whitney U test”

8. Please refer to the CONSORT checklist when preparing for your revised manuscript to ensure the comprehensiveness of information.

Answer

As asked by the reviewer, the CONSORT checklist has been completed and inserted in the revised manuscript.

Result

1. The 171 samples in the experiment group consist of only 104 using the chatbot and the remaining of 67 never use it. So, you cannot analyze the outcome using the 171. The 67 is considered as dropout and cannot be included in the final analysis. The reason is the 67 samples never access to the chatbot and considered no different than those in control group. It is either you move the 67 to control group or exclude them from the analysis. If possible, please use CONSORT flowchart for your figure 2.

2. Again, your analysis on the PCC exam results needs to consider the above point I argued.

Answer

We thank the reviewer for this relevant remark. We wondered a lot about the management of randomized students in the "assigned to play" group who had not received a login for Chatprogress because they were absent from the introductory meeting (n=42); or who had not played a single game (no connection despite a personal login, n=67). As the reviewer suggests, we considered that the 42 “assigned to play” students who had not received a login for Chatprogress should be considered no different than those in control (figure 1). Nevertheless, as for a therapeutic trial, we chose to analyze "in intention to treat" all the students who had a personal login (gamers) and to analyze in "per-protocol" all the students who had connected to the least once to play (users). The analysis with the gamers group, which shows a significant difference, seems to us to reflect a pedagogical reality insofar as the students do not all use the same tools or pedagogical resources. Indeed, a new pedagogical tool is never adopted by all students (PMID:35317712, 33795479) and, as for medical guidelines, there are several described barriers to the implementation of pedagogical tools (PMID: 36015887).

3. The difference although significant but small. You need to conduct additional statistical analysis such as effect size and minimal detectable change/minimum clinically important analysis to support your findings.

AnswerThanks for this comment. It appears difficult to analyze the effect size and the minimum clinically important difference in the context of the study. 

Nonetheless, median differences between:

- grades obtained in pulmonology for Controls and Gamers was 1.2 with a calculated Hodges-Lehman difference of 0.72 and a confidence interval of difference of 0.03 to 1.42

- average grades obtained in pooled pulmonology, cardiology and intensive care medicine for Controls and Gamers was 0.4675 with a calculated Hodges-Lehman difference of 0.4613 and a confidence interval of difference of 0.03 to 0.88

- grades obtained in pulmonology for Controls and Users was 1.2 with a calculated Hodges-Lehman difference of 0.75 and a 95% confidence interval of difference of 0.17 to 1.33

- average grades obtained in pooled pulmonology, cardiology and intensive care medicine for Controls and Users was 0.4525 with a calculated Hodges-Lehman difference of 0.41 and a 95% confidence interval of difference of 0.0425 to 0.775

95% CI data has been added in the revised manuscript.

Discussion

1. You mentioned the limitation on the low adherence, but there is no direct discussion about why there is a significant number of students never use the chatbot.

Answer

Thank you for pointing that out, we added the following in the discussion : “An explanation to students’ weak adherence to the trial could be that, in Paris Descartes, the PCC course is taught during their first clinical year. Students during that year learn to juggle between hospital obligations, lectures, study groups and self-education. That could have hindered the introduction of yet another learning tool.”

2. You mentioned the limitation on the low response rate but there is no discussion about why on the low response rate on the survey.

Low adherence to the chatbot and the survey was indeed not explained. 

Answer

Here is what we added to try to explain students’ low response rate to the survey : “Students’ poor response rate to the survey can be explained partly by the timing of when it was sent out. The email was sent at the end of the year, so 3 to 6 months after the completion of the PCC exam for two thirds of the students. It was sent out in July and students had 2 months to fill it out. It was therefore sent during the summer break, when students are usually the least responsive. Also, no reminder was sent out after the initial email. “

3. “Our results showed that using Chatprogress, a chatbot system, improved students’ results on an academic test.” I less agree with the statement as it needs to be cautiously accepted. Although significant, but the visual analysis (mean comparison) shows little difference and no significant correlation between the frequency of chatbot use with grades. I suggest changing to a more cautious tone such as “potentially improved” or “plausible in improving”.

Answer

We modified the first sentence of the discussion accordingly. 

4. More aspect on the feasibility can be discussed as the survey findings yielded several interesting issues. For example, the same game (e.g., I, III) is rated in both as like the most and least. Another example is game II and V has no participant complete rounds.

Answer

We thank the reviewer for this very interesting comment. Indeed, oursurvey provides food for thought to improve the usability and deployment of chatbots. Concerning games I and III rated in both as like the most and least, it seems to us that this discrepancy highlights the variability of profiles and sensitivities of medical students and may also reflect the difference in knowledge level between them. Concerning the two games that were never completed, there is no correlation with their level of difficulty (II was rated as easy by the editors, V as difficult). We believe that the pedagogical use of chatbots would justify debriefing sessions as for any medical simulation. These sessions would make it possible to identify the cause of the non-completion: poorly constructed scenario or medical reasoning defect that could be corrected and improved. We therefore added the following sentence in the discussion section: "As the survey shows that two games could not be completed, we think that a debriefing session would allow us to understand the difficulties either inherent to the chatbot or the difficulties of medical reasoning encountered by the students and, thus, to correct them”.

5. Another interesting aspect is high number intentionally get GO (Game Over), is it because they just want to quickly get the answer (information reading) rather than playing the game? You touched about this in brief as “learning on the platform is done through trial and error, a well-known and efficient learning method”. However, a deeper analysis is required. Is it perhaps playing game is time consuming, so they quickly want the answer? So, they can use the save time for something else? Please add more reference and argument.

6. “This suggests that students were not merely guessing their answers but were genuinely thinking the questions through”. I found this argument is weak if want to be based on high number of voluntary GO. It can be that they do not want to think at all and terminate the game to directly get answer/reasoning?

Answers to questions 5 and 6

We will here be answering questions 5 and 6. First, thank you very much for the very interesting exchange, we indeed thought of the same arguments, but some elements do back up our thought process. First, 59% of students who successfully made it through a game (meaning, no GO was encountered) played that game again. Since our chatbot modulates its pedagogical comments based the students’ answers, students understood that there is more to learn from the game than just making it to the end. A lot of the learning happens by taking different pathways of answers, with GO or playing the game again (with different answers) after completion. 

Also, even if we do consider that students get GO voluntarily to speed through the game, to get answers very quickly, and don’t follow the clinical reasoning step by step, 88.8% of students considered this format useful for learning concept. So even if the tool was misused, our ultimate goal is fulfilled: that platform-helped students learned something.

7. I am not sure if cross-over is limited. Even with 15.9% (n=27) response rate, there is an 11% answered YES that they make anyone else try the CHATPROGRESS. There is a significant non-response bias to be considered. Moreover, you cannot sure that they at least not discuss about the information with their friends (not about using the chatbot, but sharing the knowledge they gained from it).

Answer

We indeed didn’t point out the remaining possibility for cross over, thank you for pointing it out. We modified the beginning of our discussion accordingly: 

“We made sure, to the best of our ability, to limit the cross-over between groups, by first only meeting up with the students that were randomized in the trial group. Personal identification was also distributed, limiting the access of students in the control group to the tool. We also made sure the group contamination was limited by asking the question in the survey, which pointed out that cross-over was limited among those who answered the survey. However, we cannot eliminate a non-response bias, remaining ignorant of the sharing of the platform by those who did not fill out the survey. We also cannot argue that some of the knowledge acquired by using the chatbot could have been shared between peers.”

8. “Similarly, only 13% of the Gamers completed the survey”. 13% or 15.9%? as it is contradicted between the one mentioned in the discussion and the one reported in the result.

Answer

Thank you for pointing out two of our mistakes. First the mismatch between the discussion and the result, that we corrected. Second, we realized that to calculate the percentage of students that answered the survey, we can only consider the ones that actually tried the game, the Users. We will thus be changing it from percentage of Gamers to percentage of Users, bringing the percentage of students that answered the survey up to 26%.

9. You need to critically discuss that student motivation to pass exam is multifactorial, not just about using the game. Therefore, this may explain of why there is only a small difference between the experiment and control outcome. Moreover, the use of game perhaps gave an opportunity bias as the intensity is not properly controlled and similarly levelled (example, the experiment group has double the time for revision – from book and game). Biases and limitation of using RCT in medical education research need to be properly discussed.

Answer

We thank the R2 for this interesting comment. First, it is less likely for students to have double their time for revision, since those who did not have access to the chatbot could have spent that time studying from the more traditional sources. Taking into account this comment, we have added this at the end of our discussion: « In addition, we understand that students’ ability to pass an exam is multifactorial, going from students’ schedule, to participation rate, to interest in the field or simply luck on exam day. We tried to minimize the confusion biais by performing a randomized controlled trial, with however a remaining attrition bias, that is hard to take into consideration considering the limited number of students per trimester, and evaluation bias on the evaluation of the primary outcome with students of different trimesters being evaluated on different topics”. 

10. If possible, include more reference and provide a deep discussion.

Answer

Thank you very much R2 for the interesting comment, we really hope that our answers to your questions helped deepen our discussion, enrich the conversation, and solidify the essence of our paper. Indeed, a few more references need to be added to support all of the above and we did add : 

1. Pavel Smutny, Petra Schreiberova, Chatbots for learning: A review of educational chatbots for the Facebook Messenger, Computers & Education, Volume 151,2020, 103862, ISSN 0360-1315,

2. Romli, M., Wan Yunus, F., Cheema, M.S. et al. A Meta-synthesis on Technology-Based Learning Among Healthcare Students in Southeast Asia. Med.Sci.Educ. 32, 657–677 (2022). 

3. Effects of COVID-19 on Japanese medical students’ knowledge and attitudes toward e-learning in relation to performance on achievement tests. Sekine M, Watanabe M, Nojiri S, Suzuki T, Nishizaki Y, et al. (2022) Effects of COVID-19 on Japanese medical students’ knowledge and attitudes toward e-learning in relation to performance on achievement tests. PLOS ONE 17(3): e0265356. 

My condolences to your team and family of Professor Guy Meyer. I believe he will be proud with the work.

We all thank you very much for your heartfelt condolences. 

---

## [Decision Letter · Decision Letter 1]

28 Oct 2022

PONE-D-22-00159R1Chatbot-based serious games: a useful tool for training medical students? A randomized controlled trial.PLOS ONE

Dear Dr. Planquette,

Thank you for submitting your manuscript to PLOS ONE. After careful consideration, we feel that it the revised version is markedly better, but some minor issues remain to be addressed. First, in abstract "The primary aim was to demonstrate an increase in..", "Secondary aims were to demonstrate an increase in scores..". Authors are encouraged to replace "demonstrate" by another term, such as evaluate or assess, and change the rest of the sentence accordingly.Second, authors are encouraged to further discuss the low response rate following reviewer's comments

We look forward to receiving your revised manuscript.

Kind regards,

Elsayed Abdelkreem, MD, PhD

Academic Editor

PLOS ONE

Journal Requirements:

Reviewers' comments:

Reviewer's Responses to Questions

**Comments to the Author**

1. If the authors have adequately addressed your comments raised in a previous round of review and you feel that this manuscript is now acceptable for publication, you may indicate that here to bypass the “Comments to the Author” section, enter your conflict of interest statement in the “Confidential to Editor” section, and submit your "Accept" recommendation.

Reviewer #1: All comments have been addressed

Reviewer #2: All comments have been addressed

2. Is the manuscript technically sound, and do the data support the conclusions?

Reviewer #1: Yes

Reviewer #2: Yes

3. Has the statistical analysis been performed appropriately and rigorously? 

Reviewer #1: Yes

Reviewer #2: Yes

4. Have the authors made all data underlying the findings in their manuscript fully available?

Reviewer #1: Yes

Reviewer #2: Yes

5. Is the manuscript presented in an intelligible fashion and written in standard English?

Reviewer #1: Yes

Reviewer #2: Yes

6. Review Comments to the Author

Reviewer #1: (No Response)

Reviewer #2: Thank you so much for addressing my comments and I found the article has been improved.

If I may suggest, perhaps in your limitation "The relatively weak adherence of students to the trial and the survey." you need to add another point where you only did sent out the survey once and no particular strategy was implemented to encourage students to answer the questionnaire, then suggest what strategy can be used to enhance response for future research. Perhaps you can write something as "We only blast the survey once without any reminder or incentive provided. Hence, at least a subsequent reminder is required after the initial survey sent or incentive provided may increase the response rate [XX]. However, due to [your reasons] abstain us from doing such strategies."

Reference: Sammut, D. R., Griscti, D. O., & Norman, P. I. J. (2021). Strategies to improve response rates to web surveys: A literature review. International Journal of Nursing Studies, 123, 104058. doi:10.1016/j.ijnurstu.2021.104058

7. PLOS authors have the option to publish the peer review history of their article (what does this mean?). If published, this will include your full peer review and any attached files.

Reviewer #1: **Yes: **Ayman Zaky Elsamanoudy

Reviewer #2: **Yes: **Muhammad Hibatullah Romli

---

## [Author Response · Author response to Decision Letter 1]

19 Nov 2022

Response to Reviewers 

General comments from Editor

1. In abstract "The primary aim was to demonstrate an increase in..", "Secondary aims were to demonstrate an increase in scores..". Authors are encouraged to replace "demonstrate" by another term, such as evaluate or assess, and change the rest of the sentence accordingly.

Answer: Changes to the paper were made accordingly, by replacing “demonstrate” by “evaluate”. 

2. Authors are encouraged to further discuss the low response rate following reviewer's comments

Answer: with the help of Reviewer �2’s comment, we’ve made the necessary changes and added : “This is also why a second blast, a few weeks later, was not sent, although a pre-notification or a subsequent reminder after the initial survey would have increased the response rate (18). An incentive was also not considered, although it was also shown to be linked to an increased response rate, with the budget entirely dedicated to the development of the chatbot and the student’s incentive to use it. 

Reviewers' comments:

Reviewer #2:

If I may suggest, perhaps in your limitation "The relatively weak adherence of students to the trial and the survey." you need to add another point where you only did sent out the survey once and no particular strategy was implemented to encourage students to answer the questionnaire, then suggest what strategy can be used to enhance response for future research. 

Perhaps you can write something as "We only blast the survey once without any reminder or incentive provided. Hence, at least a subsequent reminder is required after the initial survey sent or incentive provided may increase the response rate [XX]. However, due to [your reasons] abstain us from doing such strategies."

Reference: Sammut, D. R., Griscti, D. O., & Norman, P. I. J. (2021). Strategies to improve response rates to web surveys: A literature review. International Journal of Nursing Studies, 123, 104058. doi:10.1016/j.ijnurstu.2021.104058

Answer: Thank you very much for all your very interesting comments thus far. We have made the necessary changes, by adding: This is also why a second blast, a few weeks later, was not sent, although a pre-notification or a subsequent reminder after the initial survey would have increased the response rate (18). An incentive was also not considered, although it was also shown to be linked to an increased response rate, with the budget entirely dedicated to the development of the chatbot and the student’s incentive to use it.”

---

## [Editor Report · Decision Letter 2]

22 Nov 2022

Chatbot-based serious games: a useful tool for training medical students? A randomized controlled trial.

PONE-D-22-00159R2

Dear Dr. Planquette,

We’re pleased to inform you that your manuscript has been judged scientifically suitable for publication and will be formally accepted for publication once it meets all outstanding technical requirements.

Kind regards,

Elsayed Abdelkreem, MD, PhD

Academic Editor

PLOS ONE
---

## [Editor Report · Acceptance letter]

1 Dec 2022

PONE-D-22-00159R2 

Chatbot-based serious games: a useful tool for training medical students? A randomized controlled trial. 

Dear Dr. Planquette:

I'm pleased to inform you that your manuscript has been deemed suitable for publication in PLOS ONE. Congratulations! Your manuscript is now with our production department. 

Kind regards, 

on behalf of

Dr. Elsayed Abdelkreem 

Academic Editor

PLOS ONE